# FedIGL: Federated Invariant Graph Learning for Non-IID Graphs

Lingren Wang[1]    Wenxuan Tu[2*]   Jiaxin Wang[3]    Xiong Wang[2]
Jieren Cheng[2*]   Jingxin Liu[3]

[1]School of Information and Communication Engineering, Hainan University
[2]School of Computer Science and Technology, Hainan University
[3]School of Cyberspace Security, Hainan University
{twx,992730}@hainanu.edu.cn

## Abstract

Federated Graph Learning (FGL) effectively facilitates cross-domain graph model training by enabling decentralized learning across multiple domains, while ensuring data privacy through local data storage and communication of model updates instead of raw data. Existing approaches usually assume shared generic knowledge (e.g., prototypes, spectral features) via aggregating local structures statistically to alleviate structural heterogeneity. However, imposing overly strict assumptions about the presumed correlation between structural features and the global objective often fails in generalizing to local tasks, leading to suboptimal performance. To tackle this issue, we propose a **Fed**erated **I**nvariant **G**raph **L**earning (**FedIGL**) framework based on invariant learning, which effectively disrupts spurious correlations and further mines the invariant factors across different distributions. Specifically, a server-side global model is trained to capture client-agnostic subgraph patterns shared across clients, whereas client-side models specialize in client-specific subgraph patterns. Subsequently, without compromising privacy, we propose a novel Bi-Gradient Regularization strategy that introduces gradient constraints to guide the model in identifying client-agnostic and client-specific subgraph patterns for better graph representations. Extensive experiments on graph-level clustering and classification tasks demonstrate the superiority of FedIGL against its competitors.

## 1   Introduction

Graph Neural Networks (GNNs) research [50, 33, 9, 11, 38, 36, 8] is rapidly growing due to the ability of GNNs to learn representations from graph-structured data. In practice, centralizing large amounts of real-world graph data for training is prohibitive due to privacy concerns and regulatory restrictions [26, 47, 35]. Federated Graph Learning (FGL), a growing distributed learning paradigm, offers a potential solution to this challenge while preserving data privacy [24, 23]. Nonetheless, the non-IID problem remains a major challenge in FGL, as graph data from different distributions usually vary significantly [40].

Existing approaches typically rely on the assumption that generic knowledge learned from training on non-IID clients can be effectively reconstructed across clients to enable collaborative training [53]. The shared knowledge includes consensus prototypes [22, 48], generic spectral knowledge [32], and structure encoder parameters [31], which are introduced as shared knowledge representations to facilitate the execution of graph-level learning tasks. Despite the enormous success, existing methods overly rely on statistical correlation, misleadingly assuming that the robust representations learned from prior knowledge are widely applicable. The correlation between generic knowledge

---

*Corresponding author.

39th Conference on Neural Information Processing Systems (NeurIPS 2025).

and the target is not necessarily task-related, and such spurious correlations embedded in the learned representations often fail to generalize in real-world scenarios. Furthermore, these approaches typically upload the learned knowledge or prototypes to the server, which may lead to potential data leakage. Therefore, it is critical to promote inter-client negotiation in the framework of federated graph learning without compromising data privacy.

An intuitive solution is to exploit factors that remain consistently stable and effective across clients to mitigate the impact of spurious correlations that merely reflect statistical commonality. In other words, the global model should be capable of identifying invariant factors across clients. Empirical observations indicate that graphs with different distributions often share common subgraph patterns [54], even when their distributions differ significantly, as shown in Fig. 1. This observation inspires the following idea: if we can identify subgraph patterns that are shared across different distributions, these common patterns can serve as a foundation for inter-client collaboration and improve the generalization of the global model. In contrast, distribution-specific patterns should be retained locally on each client to prevent them from negatively impacting the global representation. This insight prompts us to consider two fundamental questions: (1) How to discover invariant subgraph patterns across different distributions in FGL? (2) How can one extract invariant subgraph patterns in a privacy-preserving manner, considering that FL prohibits data sharing across clients? To the best of our knowledge, both questions remain largely unexplored.

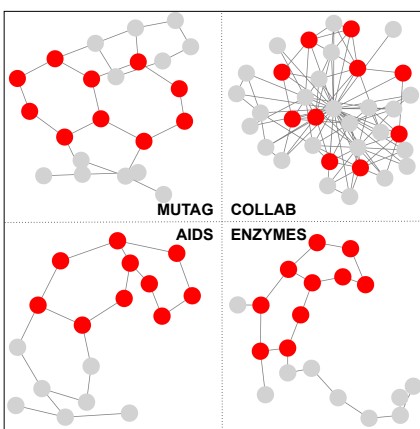

Figure 1: Illustration of shared subgraphs in different distributions in TU-Datasets [30], where the red nodes highlight the common structure.

To tackle these challenges, we propose a **Fed**erated **I**nvariant **G**raph **L**earning (FedIGL) , which aims to identify invariant subgraph patterns shared across clients while preserving local privacy. To address the first question, inspired by invariant graph learning [20, 47], which focuses on improving generalization to out-of-distribution graphs, we design a Federated Subgraph Generator (FSG) to extract client-agnostic and client-specific subgraphs. The generated client-agnostic and client-specific subgraphs are used to promote negotiation among clients and maintain the heterogeneity inherent to each client, respectively. To address the second question, we introduce a novel Bi-Gradient Regularization strategy that imposes consistency and diversity constraints on gradients. It is effectively guides the generator to learn disentangled subgraph patterns while ensuring data privacy is not compromised. After obtaining the client-specific subgraphs, we design a local model for each client that is excluded from collaborative training and is trained solely on these subgraphs. FedIGL is encouraged to discover invariant subgraph patterns across data distributions, thereby mitigating client drift caused by graph heterogeneity.

- To the best of our knowledge, this is the first work leveraging invariant learning in federated graph learning to enhance generalization under non-IID client settings.

- We propose the Bi-Gradient Regularization strategy, which can coordinate the clients to learn disentangled subgraph patterns without compromising data privacy.

- We conduct extensive experiments to verify both our theoretical results and the superiority of FedIGL, which consistently outperforms existing approaches on graph-level classification and clustering tasks.

## 2 Related Work

**Federated Graph Learning.** FGL enables distributed training of GNNs across multiple parties, facilitating collaborative learning on graph-structured data without compromising data privacy [43, 41, 10, 14, 46, 49, 19, 18, 37, 12]. Due to significant differences in client distribution and graph structure across domains, low inter-graph similarity hinders unified processing [44, 7, 13, 4, 5]. Existing methods mitigate structural heterogeneity by leveraging shared, pre-trained representations from cross-domain client models [42]. Examples include prototype-based structures [40, 53], spectral

feature alignment [32], and shared structural encoder parameters [31]. FedSSP [32] shares generalized spectral knowledge with a personalized module to adapt to client-specific graph structures, while FedGCN [22] leverages multi-source clustering to generate global consensus representations, enhancing its ability to handle complex graph structures. Despite their success, most methods rely heavily on assumed shared knowledge, which limits adaptability to diverse distributions. Since this knowledge is learned via pre-trained shared parameters unrelated to task causality, distribution shifts can degrade representation quality and harm model performance [45, 39, 17, 11].

**Invariant Graph Learning (IGL).** Invariant Learning is a class of learning methods focused on distribution generalization or robust modeling [2, 1]. Its main idea is to learn representations or predictive functions that remain stable and effective across different environments or data distributions. As previously discussed, although graphs from different distributions are heterogeneous, they share certain common subgraph patterns. Building upon these findings, IGL has emerged as a prominent research direction in recent years [20, 47, 29]. The main idea of IGL is designing a subgraph generator to partition a graph into two components: the invariant subgraph, which captures structures that are consistent across different distributions, and the environment-specific subgraph, which represents structures that are present only in particular distributions. In this paper, we extend invariant graph learning to federated learning, where each client with a distinct distribution is treated as a separate environment.

## 3   Preliminaries

**Federated Learning (FL).** Given a local dataset $(x, y) \sim P_k$ for each client $k$, where $P_k$ denotes a client-specific data distribution, the goal of standard FL approaches [27, 21] is to learn a global model that minimizes the empirical risk across all client distributions, defined as:

$$\theta^* = \arg \min_{\theta} \sum_{k=1}^{K} \mathbb{E}_{(x,y)\sim P_k} \, \ell(f_\theta(x), y), \tag{1}$$

where $\ell(\cdot)$ is a task-specific loss function and $f_\theta(\cdot)$ is the global model parameterized by $\theta$. In practice, FL methods typically decompose this global objective into a weighted sum of local empirical losses and perform independent optimization on each client. However, independently optimizing local objectives often leads to suboptimal convergence [6, 41], particularly when client distributions exhibit significant heterogeneity. This is because, under non-IID conditions, the local updates from different clients may follow divergent optimization directions [44, 25, 28, 16, 3].

## 4   Methodology

In this section, we introduce our proposed method in detail, whose framework is shown in Fig. 2. First, we define our optimization objective. Then, we present the FSG identifying client-invariant subgraph patterns. Finally, we propose the Bi-Gradient Regularization strategy for objective optimization and provide its theoretical analysis. Algorithmic details can be found in Appendix A.

### 4.1   Problem Formulation

In the federated optimization objective in Eq. (1), when client data are not identically distributed, each client's optimization direction tends to push the global model along different update trajectories [40]. This leads to gradient conflicts, hindering convergence to a globally optimal solution. To mitigate this, our goal is to guide the global model to focus on features that are shared across all client distributions, while excluding distribution-specific features from global optimization [27, 6]. This strategy helps prevent conflicting updates and promotes more stable convergence. We therefore aim to incorporate client-agnostic subgraphs, namely invariant subgraphs, into global model training to better accommodate distributional shifts [20]. In contrast, client-specific subgraphs, which capture environment-specific patterns, are processed by dedicated local models that remain discrepancies to each client. In this paper, we decouple the model into two distinct components: a global model $f_g(\cdot)$, which participates in federated aggregation, and a local model $f_c(\cdot)$, which remains client-specific.

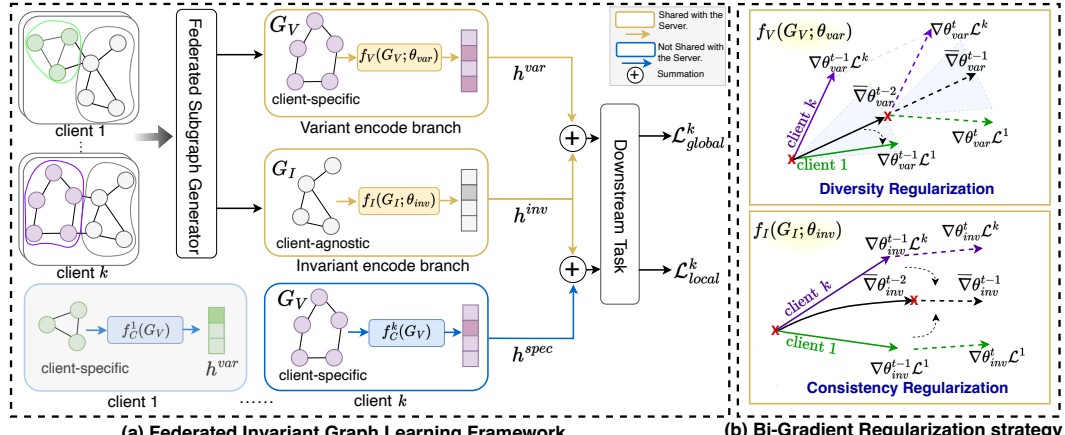

**(a) Federated Invariant Graph Learning Framework**   **(b) Bi-Gradient Regularization strategy**

Figure 2: Architecture illustration of FedIGL. The left box (a) represents the process by which clients obtain client-agnostic and client-specific subgraphs through the federated subgraph generator (FSG). The yellow and blue boxes denote parameters that are globally shared and not shared, respectively. The right box (b) shows that diversity regularization penalizes overly similar gradients in the variant encoder. Consistency regularization encourages stability and agreement in the optimization trajectory of the invariant encoder across rounds.

Based on this formulation, the optimization objective of FedIGL is defined as follows:

$$\min_{\theta_g, \{\theta_c^k\}_{k=1}^K} \sum_{k=1}^K \mathbb{E}_{(x,y)\sim P_k} \left[ \ell \left( h \left( f_g(x; \theta_g),\ f_c(x; \theta_c^k) \right),\ y \right) \right], \qquad (2)$$

where $\theta_c^k$ denotes the local model parameters of client $k$, $\ell(\cdot, \cdot)$ is the loss function, and $h(\cdot, \cdot)$ is a fusion function, such as concatenation or addition, used to integrate the outputs of the global and local models for downstream tasks. We decompose the overall optimization objective into two stages: global model optimization and local model optimization, which will be elaborated upon in the following sections.

### 4.2 Discovering Invariant and Variant Subgraphs

For the global model, we first employ a FSG to decompose each graph into client-agnostic and client-specific subgraphs. These subgraphs are then encoded separately, and their resulting representations are combined and fed into the downstream task for training.

Similar to prior work [15, 54], we implement the FSG using a graph neural network (GNN). Given graph $G$ with $n$ nodes and its adjacency matrix $\mathbf{A} = \{0, 1\}^{n \times n}$, where $\mathbf{A}_{i,j} = 1$ represents that there exists an edge between node $i$ and $j$, and $\mathbf{A}_{i,j} = 0$ otherwise. The FSG first generates a mask matrix $\mathbf{M} \in \mathbb{R}^{n \times n}$ for $\mathbf{A}$:

$$\mathbf{M}_{ij} = \text{MLP}\big(\text{CONCAT}(\mathbf{Z}_i, \mathbf{Z}_j)\big), \quad \mathbf{Z} = \text{GNN}(G), \qquad (3)$$

where $\text{MLP}(\cdot)$ is a multilayer perceptron and $\text{CONCAT}(\cdot, \cdot)$ is the concatenation operation, $\mathbf{Z}_i$ denotes the representation of the $i$-th node in graph $G$. We use MLP instead of the inner product to generate the matrix $\mathbf{M}$, since in federated learning, the inner product may leak structural information by enabling edge reconstruction [22]. Then, we can obtain the adjacency $\mathbf{A}_I$ and $\mathbf{A}_V$ corresponding to the client-agnostic subgraph $G_I$ and the client-specific subgraph $G_V$:

$$\mathbf{A}_I = Top_t(\mathbf{M} \odot \mathbf{A}), \quad \mathbf{A}_V = \mathbf{A} - \mathbf{A}_I, \qquad (4)$$

where $Top_t(\cdot)$ select the elements in $\mathbf{M}$ whose size is the top percent $t$, and $t$ is a hyperparameter. After obtaining $G_I$ and $G_V$, we adopt two branches, invariant and variant encode branch, to encode the obtained client-agnostic and client-specific subgraphs representation, respectively:

$$\mathbf{h}^{inv} = \mathcal{R}\big(f_I(G_I; \theta_{inv})\big), \quad \mathbf{h}^{var} = \mathcal{R}\big(f_V(G_V; \theta_{var})\big), \qquad (5)$$

where $\mathcal{R}(\cdot)$ is used to obtain the graph-level representation, and $f(\cdot)$ is the graph encoder. Then, we use the sum of $\mathbf{h}^{inv}$ and $\mathbf{h}^{var}$ as the representation of graph $G$ to train the global model. The global model loss in client $k$ is then defined as:

$$\mathcal{L}_{global}^k(\theta_{FSG}, \theta_{inv}, \theta_{var}) = \ell(\mathbf{h}^{inv} + \mathbf{h}^{var}, y), \qquad (6)$$

where $\theta_{FSG}$ is the parameters of the FSG.

For the client model, we define a private encoder $f_C(\cdot)$ for each client to encode the client-specific subgraph. Then, we take $G_V$ from the subgraph generator as the input of the local model, and we can get the client-specific subgraph representation in client $k$:

$$\mathbf{h}_k^{spec} = \mathcal{R}\big(f_C^k(G_V)\big). \qquad (7)$$

It should be emphasized that $\mathbf{h}^{spec}$ and $\mathbf{h}^{var}$, though derived from the same client-specific subgraph, are encoded using distinct encoders. After obtaining the client-specific subgraph representation, we use the sum of $\mathbf{h}^{inv}$ and $\mathbf{h}^{spec}$ as the overall graph representation to train the local model in client $k$. The local model loss is defined as:

$$\mathcal{L}_{local}^k = \ell(\mathbf{h}^{inv} + \mathbf{h}^{spec}, y). \qquad (8)$$

Note that we fix the global model when training the local model. That is, $\mathbf{h}^{inv}$ obtained from the global model does not participate in gradient calculation and only optimizes the local model in minimizing $\mathcal{L}_{local}$.

## 4.3 Bi-Gradient Regularization strategy

After introducing the overall FedIGL framework, we now provide detailed insights into the optimization strategies for the global and local models. In this section, we focus on how to optimize the global model to achieve disentangled and effective feature learning across distributed clients. We begin with the global optimization objective for a specific client $k$:

$$min\ \mathcal{L}_{global}^k(\theta_{FSG}, \theta_{inv}, \theta_{var}). \qquad (9)$$

In this objective, directly minimizing $\mathcal{L}_{global}^k$ for each client $k$ independently would degenerate to conventional federated learning objectives. This approach fails to enforce collaboration among clients for learning client-invariant and client-specific representations. Accordingly, we delve deeper into optimizing the global model with respect to this objective.

Assume the invariant subgraph generator is ideal, such that it can extract the same semantic subgraph pattern from different distributed graphs. Then, the feature representations of such subgraphs should be consistent across all clients. Structural and feature differences can be reflected by GNN gradients, as proved in [48]. Consequently, if the encoder $f_I$ is applied uniformly across clients, the optimization gradients with respect to $\theta_{inv}$ should also align across any clients $k, k'$:

$$\nabla_{\theta_{inv}} \mathcal{L}^k{}_{global} = \nabla_{\theta_{inv}} \mathcal{L}^{k'}{}_{global}, \qquad (10)$$

where $\nabla_{\theta_{inv}} \mathcal{L}^k{}_{global}$ represents the gradient of invariant branch encoder $f_I$ when optimizing the global loss Eq.( 6) for client $k$. Similarly, each client-specific subgraph captures features unique to each distribution. Therefore, the feature difference should drive different optimization directions for the variant branch encoder $f_V$ with the same parameters. More precisely, the gradient between any clients should satisfy:

$$\| \nabla_{\theta_{var}} \mathcal{L}^k - \nabla_{\theta_{var}} \mathcal{L}^{k'} \| \geq \varepsilon, \qquad (11)$$

where $\varepsilon$ is a predefined margin enforcing gradient diversity across clients for the variant branch. This inequality indicates that the optimization direction of heterogeneous features from different clients on the same encoder should have significant differences.

Building upon these observations, we design a novel **Bi-Gradient Regularization strategy**. The central idea is to regulate the update directions of the invariant and variant encoders, such that 1) the invariant branch encoder gradients across clients are encouraged to be consistent; 2) the variant branch encoder gradients are enforced to be diverse (repulsion beyond margin $\varepsilon$).

We achieve this by using the global aggregated gradients from the previous round as reference directions. In each round $t$, all clients' current gradients are compared with the previous global aggregation gradients, and regularization is applied accordingly. The total loss for the global model is then defined as:

$$\mathcal{L}_{total} = \mathcal{L}_{global} + \lambda \, || \nabla_{\theta_{inv}^t} \mathcal{L}_{global} - \overline{\nabla}_{\theta_{inv}^{t-1}} ||_2 + \beta \max(0, \varepsilon - || \nabla_{\theta_{var}^t} \mathcal{L}_{global} - \overline{\nabla}_{\theta_{var}^{t-1}} ||_2), \quad (12)$$

where $\lambda$ and $\beta$ are hyperparameters controlling the strength of alignment and diversity regularization, $\overline{\nabla}_{\theta_{inv}^{t-1}}$ and $\overline{\nabla}_{\theta_{var}^{t-1}}$ is the aggregated gradients with the FedAvg [27] method. The second term, **consistency regularization**, encourages stability and agreement in the invariant encoder's optimization trajectory across rounds. The third term, **diversity regularization**, penalizes overly similar gradients in the variant encoder, ensuring meaningful divergence between client-specific representations.

## 4.4 Theoretical Analysis

We provide a theoretical analysis of the FedIGL framework with invariant and variant branch encoders with bi-gradient regularization. Our objective is to demonstrate that (i) the invariant encoder achieves convergence with gradient alignment regularization; (ii) the variant encoder yields distinguishable representations across clients due to repulsive regularization.

Let $\mathcal{C} = \{1, \ldots, K\}$ be the set of clients, and $\mathcal{D}_k$ denote the local dataset of client $k$. Let $\theta = (\theta_{\text{inv}}, \theta_{\text{var}}, \theta_G)$ denote the parameters of $f_I$, $f_V$ and FSG, respectively. Define the task loss of client $k$ as $\mathcal{L}_k(\theta)$. We denote the gradients as:

$$g_k^t = \nabla_{\theta_{\text{inv}}} \mathcal{L}_k(\theta^t), \quad h_k^t = \nabla_{\theta_{\text{var}}} \mathcal{L}_k(\theta^t),$$

$$\bar{g}^{t-1} = \frac{1}{K} \sum_{k=1}^{K} g_k^{t-1}, \quad \bar{h}^{t-1} = \frac{1}{K} \sum_{k=1}^{K} h_k^{t-1}.$$

The total optimization objective at round $t$ is:

$$\min_{\theta} \sum_{k=1}^{K} \mathcal{L}_k(\theta) + \mathcal{R}_{\text{inv}} + \mathcal{R}_{\text{var}}, \quad (13)$$

where $\mathcal{R}_{\text{inv}}$ and $\mathcal{R}_{\text{var}}$ are the consistency and diversity regularization, respectively. In the federated setting, clients do not have access to the current global mean gradients $\bar{g}^t$, $\bar{h}^t$ at round $t$ when performing local updates. Therefore, we compute the regularization terms using the previous round's statistics $\bar{g}^{t-1}$, $\bar{h}^{t-1}$, which are broadcast to clients at the start of each communication round. Nevertheless, the theoretical analysis below evaluates the variance and convergence behavior with respect to the true global means $\bar{g}^t$ and $\bar{h}^t$, as is standard in federated optimization literature.

**Proposition 4.1** (Convergence of Invariant Encoder). *Assume each $\mathcal{L}_k(\theta)$ is convex and $L$-smooth, and the learning rate $\eta \leq 1/L$. Under the FEDAVG scheme with gradient alignment $\mathcal{R}_{\text{inv}}^t$, the global invariant parameters $\theta_{\text{inv}}^t$ satisfy:*

$$\min_{t=1}^{T} \mathbb{E}\left[ \left\| \nabla_{\theta_{\text{inv}}} \mathcal{L}_{global}(\theta^t) \right\|^2 \right] \leq \mathcal{O}\left( \frac{1}{T} \right). \quad (14)$$

**Proposition 4.2** (Representation Separation for Variant Encoder). *Let $h_k^t$ denote the gradient of the variant encoder for client $k$. When $\mathcal{R}_{\text{var}}^t$ is minimized, then for all $k \neq k'$, the representations remain sufficiently separated:*

$$\|h_k^t - h_{k'}^t\| \geq \epsilon. \quad (15)$$

*This guarantees per-client distinguishability in the variant branch.*

**Theorem 4.1** (Global Objective Convergence with Gradient Regularization). *Let the total loss be $\mathcal{L}_{global}(\theta) + \mathcal{R}_{inv} + \mathcal{R}_{var}$. If each $\mathcal{L}_k(\theta)$ is convex and $L$-smooth, and $\eta = \mathcal{O}(1/\sqrt{T})$, then:*

$$\min_{t=1}^{T} \mathbb{E}\left[ \left\| \nabla \mathcal{L}_{global}(\theta^t) \right\|^2 \right] \leq \mathcal{O}\left( \frac{1}{\sqrt{T}} \right). \quad (16)$$

*Indicating convergence. Furthermore, the learned representation satisfies: (i) invariant alignment across clients in $\theta_{inv}$, and (ii) per-client diversity in $\theta_{var}$.*

Based on the above formulation, we provide a theoretical justification for the effectiveness of our Bi-gradient regularization strategy. Specifically, Proposition 4.1 shows that the gradient alignment regularization term $\mathcal{R}_{\text{inv}}$ reduces the variance of client gradients in the invariant encoder, thus promoting convergence in federated optimization. Proposition 4.2 demonstrates that the repulsive regularization $\mathcal{R}_{\text{var}}$ enforces diversity among the variant gradients, enabling the model to capture client-specific characteristics. Together, these results support the disentanglement of invariant and variant subgraph patterns in a federated setting. Theorem 4.1 further establishes the convergence guarantee of our overall optimization procedure under bi-gradient regularization, quantifying the trade-off between alignment and diversity in gradient space. The proof of the above proposition and theorem see Appendix B.

# 5 Experiments

In this section, we conduct extensive experiments on graph-level classification and clustering tasks in various cross-dataset and cross-domain scenarios to validate the superiority of FedIGL. The following research questions need to be validated. **(RQ1)** Can FedIGL achieve better performance compared to SOTA baselines? **(RQ2)** Does FedIGL converge under the constraints of bi-gradient optimization? **(RQ3)** How does each of the strategies we propose contribute to the final performance? **(RQ4)** How about the hyperparameter sensitivity of FedIGL?

## 5.1 Experiment Setup

**Benchmark Datasets.** We employed a total of 19 diverse datasets across multiple domains to conduct comprehensive evaluations on both classification and clustering tasks. These domains include Small Molecules (e.g., MUTAG, BZR, COX2, DHFR, PTC_MR, AIDS, BZR_MD, and NCI1), Bioinformatics (e.g., DD, PROTEINS, OHSU, and Peking_1), Synthetic (SYNTHETIC), Social Networks (e.g., COLLAB, IMDBMULTI, and IMDB-BINARY), and Computer Vision (e.g., Letter-high, Letter-low, and Letter-med). Regarding classification tasks, We follow the settings in [32], which include six distinct experimental designs: (1) cross-dataset setting utilizing seven small molecule datasets (SM), and (2)-(6) settings that incorporate both cross-dataset and cross-domain aspects, based on datasets from two different domains (BIO-SM, SM-CV) and three different domains (BIO-SM-SN, BIO-SN-CV, SM-SN-CV). For clustering tasks, we adopt the protocols in [22], including five types of non-IID settings: (1) 2 clusters within the same domain (SM), (2) 3 clusters within the same domain (SN), (3) 15 clusters within the same domain (CV), (4) 2 clusters across two domains (SM-BIO), and (5) 2 clusters across three domains (SM-BIO-SY). The dataset and experimental implementation details are provided in Appendix C.1.

**Baseline Methods.** In both classification and clustering tasks, we compare FedIGL with two classical federated learning methods, FedAvg [27] and FedProx [21]. Additionally, we include four state-of-the-art federated graph learning methods: FedSage [52], GCFL [48], FedStar [31], and FedSSP [32]. For clustering tasks specifically, we also compare with FedGCN [22].

**Implementation Details.** To ensure fair comparisons, all methods, including FedIGL and baselines, were implemented in PyTorch and executed on the same NVIDIA GeForce RTX 3090 GPU. For graph-level structure embeddings, we use a three-layer Graph Isomorphism Network (GIN) [51] with a hidden dimension of 64 and batch size of 128 [34]. Model optimization is performed using the Adam optimizer with a learning rate of 1e-3. Dropout is set to 0.5 and weight decay to 5e-4 to improve generalization.

## 5.2 Experimental Results

**Performance Comparison (RQ1) .** Tab. 1 and Tab. 2 present a comparison of the performance of FedIGL against SOTA methods on graph-level classification and clustering tasks. In the classification task, FedIGL achieves the best results in 5 out of 6 classification settings and ranks second in the remaining one, demonstrating the most robust overall performance. Notably, under the single-domain SM setting, FedIGL improves over the FedSSP about a 4.3% relative gain. Moreover, although FedSSP slightly outperforms FedIGL on SM-CV, FedIGL shows more pronounced advantages in the more challenging multi-domain scenarios, indicating better generalization under stronger distribution shifts. In the clustering task, FedIGL ranks among the top methods across the five clustering settings,

demonstrating more stable clustering quality under non-IID setting. Notably, on SN and CV, FedGCN achieves a slight success on specific tasks but remains suggesting that the two methods emphasize cluster alignment consistency and clustering decision correctness, respectively. Existing methods often perform well on specific tasks but remain vulnerable to spurious correlations, which limits their generalization.

Table 1: Comparison with state-of-the-art methods on one cross-dataset and five cross-domain settings for classification tasks. The best is marked with **boldface** and the second best is with underline.

| Methods | Single-domain SM | Double-domain BIO-SM | SM-CV | Multi-Domain BIO-SM-SN | BIO-SN-CV | SM-SN-CV |
|---|---|---|---|---|---|---|
| FedAvg (ASTAT17) | $74.12 \pm 2.10$ | $67.82 \pm 1.63$ | $81.21 \pm 1.00$ | $67.31 \pm 2.56$ | $70.93 \pm 2.91$ | $75.33 \pm 1.06$ |
| FedProx (arXiv18) | $69.35 \pm 3.36$ | $67.27 \pm 4.17$ | $70.02 \pm 2.27$ | $63.89 \pm 4.33$ | $69.32 \pm 1.75$ | $67.15 \pm 2.25$ |
| FedSage (NeurIPS21) | $75.61 \pm 1.16$ | $72.60 \pm 3.18$ | $76.23 \pm 0.49$ | $70.84 \pm 0.88$ | $69.69 \pm 1.11$ | $73.36 \pm 0.86$ |
| GCFL (NeurIPS21) | $77.71 \pm 1.53$ | $72.05 \pm 2.20$ | $72.64 \pm 0.71$ | $70.43 \pm 1.39$ | $67.91 \pm 2.15$ | $71.79 \pm 0.21$ |
| FedStar (AAAI23) | $78.63 \pm 2.11$ | $72.71 \pm 1.22$ | $78.84 \pm 1.07$ | $72.60 \pm 2.45$ | $69.51 \pm 0.84$ | $75.94 \pm 0.40$ |
| FedSSP (NeurIPS24) | $\underline{79.62 \pm 2.23}$ | $\underline{73.66 \pm 2.34}$ | $\mathbf{84.29 \pm 0.68}$ | $\underline{72.37 \pm 2.18}$ | $\underline{75.07 \pm 2.70}$ | $\underline{79.12 \pm 1.23}$ |
| FedIGL(ours) | $\mathbf{83.07 \pm 1.76}$ | $\mathbf{77.02 \pm 1.32}$ | $\underline{83.14 \pm 0.28}$ | $\mathbf{75.25 \pm 1.13}$ | $\mathbf{78.50 \pm 0.44}$ | $\mathbf{79.23 \pm 1.28}$ |

Table 2: Comparison with state-of-the-art methods on three cross-dataset and two cross-domain settings for clustering tasks. Please note that the FGL methods marked with the symbol [*] have been adapted from classification to clustering tasks, as was done in previous studies. The best is marked with **boldface** and the second best is with underline.

| Domain | Metric | FedAvg ASTAT17 | FedProx arXiv18 | FedSage[*] NeurIPS21 | GCFL[*] NeurIPS21 | FedStar[*] AAAI23 | FedSSP[*] NeurIPS24 | FedGCN AAAI25 | FedIGL Ours |
|---|---|---|---|---|---|---|---|---|---|
| SM | ACC | $35.3 \pm 1.1$ | $69.4 \pm 3.4$ | $55.6 \pm 1.4$ | $61.1 \pm 1.8$ | $58.9 \pm 2.4$ | $\underline{76.4 \pm 0.5}$ | $75.9 \pm 0.8$ | $\mathbf{77.0 \pm 1.1}$ |
|  | NMI | $10.2 \pm 1.5$ | $8.4 \pm 2.9$ | $12.2 \pm 1.3$ | $8.7 \pm 2.4$ | $12.0 \pm 1.2$ | $12.9 \pm 3.0$ | $\underline{24.9 \pm 3.0}$ | $\mathbf{25.7 \pm 1.5}$ |
|  | ARI | $8.4 \pm 0.9$ | $9.5 \pm 2.1$ | $7.6 \pm 0.6$ | $9.4 \pm 2.4$ | $0.1 \pm 0.8$ | $\underline{34.1 \pm 2.3}$ | $31.1 \pm 3.4$ | $\mathbf{36.3 \pm 1.4}$ |
|  | F1 | $52.7 \pm 1.3$ | $48.2 \pm 2.5$ | $50.2 \pm 1.0$ | $43.3 \pm 1.6$ | $49.7 \pm 2.8$ | $\underline{68.2 \pm 2.5}$ | $67.1 \pm 1.5$ | $\mathbf{69.4 \pm 2.5}$ |
| SN | ACC | $61.5 \pm 2.2$ | $71.4 \pm 2.8$ | $53.3 \pm 1.9$ | $52.1 \pm 2.3$ | $51.7 \pm 2.7$ | $\underline{73.6 \pm 2.3}$ | $66.6 \pm 2.3$ | $\mathbf{73.7 \pm 0.3}$ |
|  | NMI | $15.6 \pm 1.7$ | $11.6 \pm 2.0$ | $14.8 \pm 1.4$ | $12.5 \pm 2.3$ | $13.7 \pm 2.8$ | $11.8 \pm 3.4$ | $\mathbf{30.4 \pm 6.6}$ | $\underline{23.1 \pm 1.2}$ |
|  | ARI | $10.3 \pm 1.2$ | $12.3 \pm 2.7$ | $11.6 \pm 2.8$ | $13.2 \pm 2.3$ | $12.4 \pm 1.9$ | $31.9 \pm 1.7$ | $\mathbf{34.1 \pm 5.3}$ | $\underline{32.6 \pm 2.1}$ |
|  | F1 | $58.1 \pm 2.8$ | $53.7 \pm 2.2$ | $49.3 \pm 2.0$ | $52.3 \pm 1.6$ | $50.7 \pm 2.3$ | $\mathbf{64.9 \pm 2.6}$ | $50.7 \pm 2.4$ | $\underline{62.7 \pm 2.7}$ |
| CV | ACC | $33.8 \pm 0.7$ | $\underline{38.6 \pm 2.1}$ | $10.1 \pm 1.4$ | $13.7 \pm 2.0$ | $12.4 \pm 2.7$ | $34.2 \pm 2.6$ | $34.6 \pm 2.8$ | $\mathbf{39.2 \pm 1.1}$ |
|  | NMI | $15.9 \pm 2.3$ | $12.1 \pm 2.7$ | $\underline{30.5 \pm 1.7}$ | $17.7 \pm 2.4$ | $22.4 \pm 2.5$ | $12.0 \pm 2.4$ | $\mathbf{34.2 \pm 1.4}$ | $27.3 \pm 1.8$ |
|  | ARI | $10.5 \pm 2.0$ | $12.7 \pm 2.1$ | $13.6 \pm 1.8$ | $14.3 \pm 2.7$ | $15.3 \pm 2.1$ | $\underline{33.1 \pm 3.3}$ | $19.3 \pm 1.8$ | $\mathbf{38.1 \pm 2.6}$ |
|  | F1 | $49.7 \pm 1.5$ | $54.5 \pm 2.5$ | $10.4 \pm 1.7$ | $13.2 \pm 1.4$ | $11.6 \pm 1.9$ | $\underline{35.3 \pm 1.5}$ | $31.6 \pm 3.1$ | $\mathbf{36.2 \pm 2.4}$ |
| SM-BIO | ACC | $54.3 \pm 2.9$ | $67.1 \pm 1.9$ | $57.4 \pm 2.2$ | $60.1 \pm 1.8$ | $59.5 \pm 1.6$ | $\underline{72.3 \pm 2.1}$ | $69.2 \pm 0.6$ | $\mathbf{73.6 \pm 1.0}$ |
|  | NMI | $11.8 \pm 2.1$ | $7.9 \pm 2.3$ | $5.2 \pm 2.1$ | $4.7 \pm 2.4$ | $5.3 \pm 1.6$ | $11.3 \pm 1.2$ | $\underline{14.0 \pm 2.7}$ | $\mathbf{23.0 \pm 1.6}$ |
|  | ARI | $7.2 \pm 3.0$ | $8.7 \pm 1.6$ | $4.2 \pm 2.7$ | $3.2 \pm 2.3$ | $3.8 \pm 2.0$ | $\mathbf{32.4 \pm 2.1}$ | $17.5 \pm 3.1$ | $\underline{30.7 \pm 2.5}$ |
|  | F1 | $48.7 \pm 2.4$ | $44.3 \pm 2.5$ | $49.9 \pm 0.5$ | $47.3 \pm 1.5$ | $51.7 \pm 2.2$ | $\mathbf{63.7 \pm 1.8}$ | $59.1 \pm 0.9$ | $\underline{59.4 \pm 3.8}$ |
| SM-BIO-SY | ACC | $56.1 \pm 2.8$ | $68.3 \pm 1.1$ | $57.6 \pm 1.9$ | $59.1 \pm 2.0$ | $57.9 \pm 2.6$ | $\underline{75.9 \pm 0.4}$ | $68.6 \pm 1.3$ | $\mathbf{76.7 \pm 2.2}$ |
|  | NMI | $12.5 \pm 1.4$ | $8.0 \pm 2.2$ | $\underline{20.6 \pm 1.9}$ | $14.4 \pm 2.2$ | $15.7 \pm 2.4$ | $13.9 \pm 1.4$ | $13.5 \pm 2.1$ | $\mathbf{22.7 \pm 1.8}$ |
|  | ARI | $7.8 \pm 0.9$ | $8.9 \pm 2.1$ | $17.6 \pm 2.4$ | $13.7 \pm 2.8$ | $16.1 \pm 3.0$ | $\mathbf{34.8 \pm 2.6}$ | $17.2 \pm 3.6$ | $\underline{34.6 \pm 1.6}$ |
|  | F1 | $52.1 \pm 1.7$ | $46.7 \pm 2.0$ | $49.4 \pm 1.7$ | $52.3 \pm 1.9$ | $52.3 \pm 2.2$ | $\mathbf{69.7 \pm 2.3}$ | $59.4 \pm 3.8$ | $\underline{67.5 \pm 1.9}$ |

**Convergence Analysis (RQ2) .** We visualize the graph classification testing loss with respect to the communication rounds to show the convergence of FedIGL clients under the non-IID setting, as shown in Fig. 3. Despite the inherent heterogeneity of data across clients, our method demonstrates a smooth and monotonic decrease in the global objective over communication rounds. This behavior aligns with the theoretical guarantees of federated optimization. In particular, our introduction of bi-gradient regularization further stabilizes the learning dynamics by mitigating the divergence caused by client drift, leading to faster and more consistent convergence. More non-IID settings in Appendix C.4.

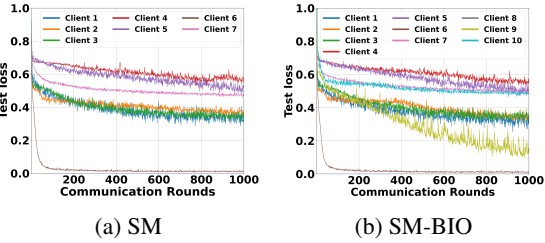

(a) SM      (b) SM-BIO

Figure 3: The loss curves of each clisnt on the SM and SM-BIO of non-IID setting.

**Ablation Study (RQ3) .** We perform ablation studies to assess how the proposed Bi-Gradient Regularization strategy contributes to the overall performance. Tab. 3 reports results of FedIGL and its variants under non-IID settings: (i) removing both consistency regularization (CR) and diversity regularization (DR), (ii) enabling only one of them (CR or DR), and (iii) enabling both. An intuitive observation is that FedIGL performs best when CR and DR are enabled simultaneously. Each component on its

Table 3: The ablation study covers both classification and clustering tasks. A checkmark (✓) indicates inclusion of the strategy, while a cross (✗) indicates its exclusion. Our non-IID settings include single-domain, double-domain, and multi-domain scenarios, corresponding to the SM, SM-BIO, SM-BIO-SN for the classification task and SM-BIO-SY for the clustering task, respectively.

| CR | DR | SM | | SM-BIO | | SM-BIO-SN (SY) | |
|----|----|----------------|------------|----------------|------------|----------------|------------|
| | | Classification | Clustering | Classification | Clustering | Classification | Clustering |
| ✗ | ✗ | 78.21 | 72.45 | 74.67 | 69.73 | 70.66 | 68.33 |
| ✓ | ✗ | 79.84 | 73.04 | 75.83 | 70.34 | 71.24 | 68.36 |
| ✗ | ✓ | 80.55 | 75.39 | 76.01 | 71.28 | 72.61 | 70.87 |
| ✓ | ✓ | **83.07** | **77.04** | **77.02** | **73.71** | **75.25** | **76.71** |

own still delivers consistent gains across most datasets. On the SM-BIO-SY clustering benchmark, introducing CR alone leads to negligible gains over the baseline. In contrast, enabling DR alone yields a noticeable improvement of 2.54%. This indicates that under multi-domain shift, encouraging representation diversity better mitigates domain bias and leads to more stable clustering. Overall, the empirical trends align with our theoretical analysis, supporting the design of jointly integrating CR and DR. Additional results are provided in Appendix C.2.

**Hyper-Parameter Study (RQ4) .** We investigate the sensitivity of several hyper-parameters in our method, including the invariance regularization strength $\lambda$, variance regularization strength $\beta$, divergence parameter $\varepsilon$, and invariant subgraph ratio $\tau$. Under the SM dataset setting, we evaluate the performance of FedIGL across various hyperparameter combinations on classification and clustering tasks, as shown in Fig. 4. The results indicate that the optimal values of $\lambda$, $\beta$, and $\varepsilon$ differ by task. For classification, $\lambda = 0.05$, $\beta = 0.2$, and $\varepsilon = 0.1$ provide the best performance balance; for clustering, $\lambda = 0.2$, $\beta = 0.25$, and $\varepsilon = 0.15$ perform best. This likely reflects distinct requirements for graph representations, suggesting that client subgraph invariance and variability are influenced by downstream tasks. The parameter $\tau$ governs the proportion of invariant subgraphs; a large $\tau$ may include excessive variant struc-

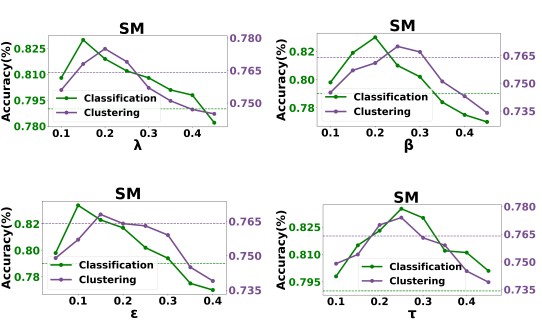

Figure 4: Hyperparameter Sensitivity analysis on SM setting. The x-axis represents the four hyperparameters: $\lambda$, $\beta$, $\varepsilon$, and $\tau$, with the left and right y-axis representing the classification and clustering accuracy, respectively.

tures, while a small $\tau$ may limit structural capture. In our experiments, $\tau = 0.25$ balances shared and client-specific structures effectively in both tasks. Additionally, we conduct further hyper-parameter sensitivity analyses under non-IID settings in Appendix C.3.

## 6    Conclusion

In this work, we present a novel Federated Invariant Graph Learning framework from a fresh perspective, aimed at capturing invariant subgraph structures to mitigate client distribution shifts. We propose a Bi-Gradient Regularization strategy applying consistency regularization to the invariant subgraph encoder and diversity regularization to the variant one, which enhances graph representation quality, stability and model performance. Overall, as a pioneering study, FedIGL provides valuable insights for addressing the graph structural differences associated with client distributional heterogeneity and is supported by extensive experimental and theoretical analysis. While our approach intuitively protects client privacy by avoiding the sharing of prototype structures, future work will further explore stringent measures for model privacy preservation.

## Acknowledgments

This work was supported by the National Natural Science Foundation of China (NSFC) (Grant No. 62562026, 62506102), the Key Research and Development Program of Hainan Province (Grant No. ZDYF2024GXJS014, ZDYF2023GXJS163), the Hainan Province Graduate Innovation Research Project (No. Qhyb2023-104), and the Natural Science Foundation of Hainan University (Grant No. XJ2400009401).

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

# A    Algorithm

---

**Algorithm 1** Optimization process of FedIGL

---

**Input:** Maximum epoch $T$; Number of clients $K$; The distributed Non-IID datasets $\{\mathcal{D}_k\}_{k=1}^K$; hyper-parameters $t$, $\lambda$, $\beta$, $\varepsilon$.
**Output:** Model trained with FedIGL.
Initialize $\{\theta_{FSG}, \theta_{inv}, \theta_{var}\}$.
**for** $t = 1$ **to** $T$ **do**
    **for** $k = 1$ **to** $K$ **do**
        Obtain $G_I, G_V$ for each graph $G \in \mathcal{D}_k$ with $t$ by Eq. (4).
        Obtain subgraph representation $\mathbf{h}^{inv}, \mathbf{h}^{var}$ for $G_I, G_V$.
        Calculate $\mathcal{L}_{global}^k$ for client $k$ with $\lambda$, $\beta$, $\varepsilon$ by Eq. (12).
        Update $\{\theta_{FSG}, \theta_{inv}, \theta_{var}\}$ by Stochastic Gradient Descent.
        Fix the parameters of $\{\theta_{FSG}, \theta_{inv}, \theta_{var}\}$.
        Obtain client-specific subgraph representation $\mathbf{h}^{spec}$ with $G_V$.
        Calculate $\mathcal{L}_{local}^k$ for client $k$ by Eq. (8).
        Update local model $f_C^k$ for client $k$ by Stochastic Gradient Descent.
    **end for**
    Aggregate $\{\theta_{FSG}, \theta_{inv}, \theta_{var}\}$.
**end for**

---

# B    Proofs in Section 4.4

**Lemma B.1** (Gradient Variance Reduction for Invariant Branch). *Suppose each $L_k(\theta)$ is $L$-smooth and the initial variance of local invariant gradients satisfies $\mathbb{E}\|g_k^t - \bar{g}^t\|^2 \le \sigma^2$. Then, applying the regularization $\mathcal{R}_{inv}$ leads to exponential decay in gradient variance:*

$$\mathbb{E}\|g_k^t - \bar{g}^t\|^2 \le (1 - \lambda_1 \eta)^t \sigma^2,$$

*assuming $0 < \eta \le 1/L$ and $\lambda_1 > 0$.*

*Proof.* Using $L$-smoothness, we apply the standard gradient descent update:

$$g_k^t = g_k^{t-1} - \eta \nabla_{\theta_{inv}}^2 \mathcal{L}_k(\theta^{t-1}) + \mathcal{O}(\eta^2).$$

Applying the regularization $\mathcal{R}_{inv}$ effectively forces each $g_k^t$ to align with the global average $\bar{g}^{t-1}$, thereby reducing the variance. The gradient variance evolves as:

$$\mathbb{E}\|g_k^t - \bar{g}^t\|^2 \le (1 - \lambda_1 \eta)^t \sigma^2.$$

$\square$

## B.1    Proofs of Proposition 4.1

*Proof.* With convexity and smoothness, we apply standard convergence results for Stochastic Gradient Descent (SGD) with variance reduction. By Lemma B.1, the variance of $g_k^t$ decreases over time, which helps stabilize the FedAvg updates. Using the descent lemma and unbiased gradients, we have:

$$\mathcal{L}_{\text{global}}(\theta^{t+1}) \le \mathcal{L}_{\text{global}}(\theta^t) - \eta \left\| \nabla_{\theta_{inv}} \mathcal{L}_{\text{global}}(\theta^t) \right\|^2 + \eta^2 \mathcal{L} \sigma^2.$$

Averaging over $T$ rounds gives the $\mathcal{O}(1/T)$ rate. $\square$

## B.2    Proofs of Proposition 4.2

*Proof.* The penalty $\max(0, \epsilon - \|\bar{h}^{t-1} - h_k^t\|)^2$ pushes each $h_k^t$ to be at least $\epsilon$ away from the mean. If $\|\bar{h}^{t-1} - h_k^t\| < \epsilon$, the penalty is active, increasing the loss. At the optimum, this penalty is zero, implying that $\|\bar{h}^{t-1} - h_k^t\| \ge \epsilon$, assuming that the mean of all clients' gradients remains close to the previous gradients. And thus pairwise $\|h_k^t - h_{k'}^t\| \ge \epsilon$. Consequently, the absolute difference between the gradients of any two clients exceeds the penalty term. $\square$

## B.3 Proofs of Theorem 4.1

*Proof.* The total loss includes smooth convex functions and squared penalties. Using standard convergence bounds for smooth objectives with gradient regularization and a diminishing step size $\eta = \mathcal{O}(1/\sqrt{T})$, we get the convergence rate of $\mathcal{O}(1/\sqrt{T})$ for gradient norms. The variance reduction and margin-enforcing terms ensure stable updates for both branches. $\square$

**Response: On Computational Complexity.** The per-round per-client computation in FedIGL mainly involves two parts: 1. The **Federated Subgraph Generator (FSG)**, which consists of a $L_1$-layer GNN with complexity $O\big(L_1(|E|d + |V|d^2)\big)$ to encode node features, and an edge-wise MLP scorer with complexity $O(|E|d^2)$ for selecting invariant edges; 2. The **dual-branch GNN encoder**, each branch having $L_2$ layers, resulting in a total complexity of $O\big(L_2(|E|d + |V|d^2)\big)$. Here $|V|$ and $|E|$ denote the number of nodes and edges in the client's local graph, respectively; $d$ is the feature dimensionality of each node; and $L_1, L_2$ represent the number of GNN layers in the Federated Subgraph Generator and dual-branch encoder, respectively. Hence, the overall per-client cost per round is: $O\big((L_1 + L_2)(|E|d + |V|d^2) + |E|d^2\big)$, which remains linear in the graph size and thus comparable to standard GNN-based federated learning methods. The extra overhead from edge scoring is lightweight and only applied once per round.

# C Additional Experiments

## C.1 Experiment Dataset

**Evaluation Metrics** In classification tasks, we employ Accuracy (ACC) to assess the performance of the method. Regarding clustering tasks, we utilize widely-adopted clustering result evaluation metrics, namely Accuracy (ACC), Adjusted Rand Index (ARI), Normalized Mutual Information (NMI), and F1 Score (F1). These metrics provide multi-faceted evaluations of the clustering results. Specifically, larger values of these metrics correspond to better performance. They imply more efficient data partitioning and a more accurate capture of the underlying data structure, thereby demonstrating the superiority of the clustering method in organizing data and uncovering its inherent characteristics.

Table 4: A superscript "1" in the upper-right corner indicates that the dataset is only used for classification tasks, "2" indicates that the dataset is only used for clustering tasks, and the absence of a superscript indicates that the dataset is used for both classification and clustering tasks.

| Datasets | Domain | Classes | Graphs | A.Nodes | A.Edges |
|---|---|---|---|---|---|
| MUTAG | | 2 | 188 | 17.93 | 19.79 |
| BZR | | 2 | 405 | 35.75 | 38.36 |
| COX2 | | 2 | 467 | 41.22 | 43.45 |
| DHFR | SM | 2 | 756 | 42.43 | 44.54 |
| PTC_MR | | 2 | 344 | 14.29 | 14.69 |
| AIDS | | 2 | 2000 | 15.69 | 16.20 |
| NCI1[1] | | 2 | 4110 | 29.87 | 32.30 |
| BZR_MD[2] | | 2 | 306 | 21.30 | 225.06 |
| DD[2] | | 2 | 1178 | 284.32 | 715.66 |
| PROTEINS | BIO | 2 | 1113 | 39.06 | 72.82 |
| OHSU[1] | | 2 | 79 | 82.01 | 199.66 |
| Peking_1[1] | | 2 | 85 | 39.31 | 77.35 |
| SYNTHETIC[2] | SY | 2 | 300 | 100.00 | 196.00 |
| COLLAB[2] | | 3 | 5000 | 74.49 | 2457.78 |
| IMDB-MULTI | SN | 3 | 1500 | 13.00 | 65.94 |
| IMDB-BINARY | | 2 | 1000 | 19.77 | 96.53 |
| Letter-high | | 15 | 2250 | 4.67 | 4.50 |
| Letter-low | CV | 15 | 2250 | 4.68 | 3.13 |
| Letter-med | | 15 | 2250 | 4.67 | 3.21 |

## C.2 Ablation Study

We present additional ablation experiments across multiple non-IID settings, as shown in Tab.5. The results demonstrate that the combination of two optimization strategies significantly outperforms the use of individual strategies, thereby validating the effectiveness of our proposed optimization framework. Notably, our method not only excels in specific settings but also exhibits consistent performance across a wide range of scenarios, highlighting its robustness and adaptability to varying conditions. The proposed approach effectively handles diverse datasets and domain configurations, yielding high-quality graph representations that deliver superior performance in the classification tasks.

Table 5: Ablation study of key components, namely Consistency Regularization (CR) and Diversity Regularization (DR), of FedIGL on double-domain and multi-domain settings (SM-CV, BIO-SN-CV and SM-SN-CV) in the classification.

| CR | DR | SM-CV | BIO-SN-CV | SM-SN-CV |
|---|---|---|---|---|
| ✗ | ✗ | 78.35 | 73.89 | 69.82 |
| ✓ | ✗ | 79.76 | 74.95 | 72.31 |
| ✗ | ✓ | 80.43 | 75.11 | 71.69 |
| ✓ | ✓ | **83.14** | **78.50** | **79.23** |

## C.3 Hyper-Parameter Study

We investigate the sensitivity of several hyperparameters in our method, including the invariance regularization strength $\lambda$, the variance regularization strength $\beta$, the divergence parameter $\varepsilon$, and the invariant subgraph ratio $\tau$. The hyperparameter tuning results across the non-IID settings are presented in Fig.5, with the following key observations:(1) The value of $\tau$ is primarily influenced by cross-domain distribution shifts, rather than downstream tasks, with the optimal range identified between [0.2, 0.3]. (2) The value of $\lambda$ is notably task-dependent. For clustering tasks, the optimal range lies within [0.2, 0.3], while for classification tasks, it is within [0.1, 0.2]. (3) The optimal values for $\beta$ and $\varepsilon$ are found within the ranges of [0.2, 0.3] and [0.1, 0.2], respectively. These results provide crucial theoretical insights into hyperparameter optimization and serve as a strong foundation for adapting our model to heterogeneous data distributions in real-world applications.

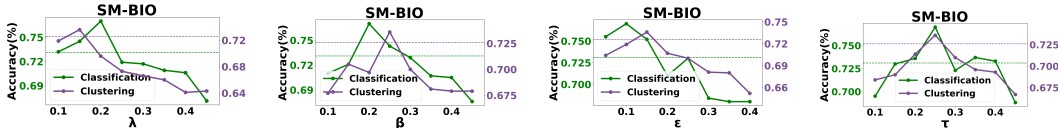

Figure 5: Hyperparameter Sensitivity analysis. The x-axis represents the four hyperparameters: $\lambda$, $\beta$, $\varepsilon$, and $\tau$, with the left and right y-axis representing the classification and clustering accuracy, respectively. The dashed lines in the figure represent the highest test accuracy of the baseline method under the settings of the SM-BIO.

## C.4 Convergence Analysis

Fig. 6 provides additional information on the relationship between graph classification loss and communication rounds in three non-IID settings. The experimental results indicate that, as the communication rounds progress, each client's loss curve exhibits a smooth, consistent decline, substantiating the effectiveness of our method in promoting model convergence. Further convergence experiments under the non-IID setting reinforce the superiority of the proposed bi-gradient regularization strategy in cross-dataset and cross-domain scenarios, demonstrating enhanced training stability and stronger generalization capability.

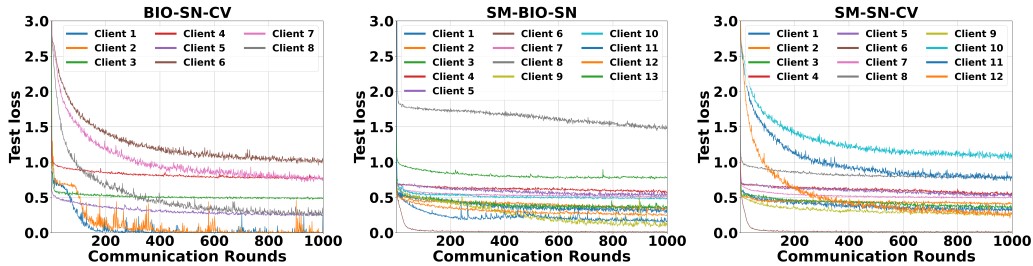

Figure 6: Loss trends of individual clients under three dataset settings: BIO-SN-CV, SM-BIO-SN, and SM-SN-CV.

