# OpenReview forum: "FedIGL: Federated Invariant Graph Learning for Non-IID Graphs"
_NeurIPS.cc/2025/Conference — NeurIPS 2025 poster_

### Official Review · Reviewer_hKG3 · 2025-06-26

**Clarity:** 4
**Significance:** 2
**Originality:** 3
**Rating:** 4
**Confidence:** 5

**Summary:**

This work focuses on federated invariant graph learning for non-IID graphs. Unlike traditional federated graph learning approaches, the proposed FedIGL incorporates invariant learning to mitigate structural heterogeneity across clients. The Bi-Gradient Regularization strategy is designed to optimize graph representations, thereby improving performance on clustering and classification tasks.

**Questions:**

a. Some formula symbols are not clearly defined. For example, the meanings of f_V and f_I in Eq. (5) are insufficiently explained and require further clarification.
b. How dose Eq. (16) contributes to the analysis of global and local convergence behavior within the proposed framework? A more detailed explanation would strengthen the theoretical analysis.
c. How is the framework implemented in scenarios where clients possess different numbers of label categories in downstream classification tasks?
d. Eq. (1) lacks appropriate citations.

**Ethical Concerns:**

["NO or VERY MINOR ethics concerns only"]

**Final Justification:**

After reading the author's response, most of my concerns have been well addressed. I maintain my score.

**Limitations:**

Please see the weaknesses and questions.

**Quality:**

3

**Strengths And Weaknesses:**

Pros:
a. The paper investigates an important yet challenging research topic in the domain of federated graph learning. The framework design is clear and well-presented.
b. The authors propose a novel federated graph embedding method that alleviates the data heterogeneity issue from the perspective of invariant learning.
c. Both theoretical analysis and empirical results demonstrate the effectiveness and superiority of the proposed method compared to advanced baseline methods.

Cons:
a. In Section 3.2, FSG is employed to identify invariant and variant subgraphs. However, it remains unclear how the subgraph generator distinguishes invariant subgraphs, and what mechanism are in place to ensures the effectiveness of this process？
b. The manuscript lacks sufficient clarity regarding implementation details, particularly in downstream scenarios where the number of label categories may vary across clients.
c. The time complexity of the proposed method is not thoroughly analyzed, and there is limited discussion on computational efficiency. This makes it somewhat difficult to assess the scalability of the approach, especially in large-scale settings.

---

> ### Author Rebuttal · Authors · 2025-07-28
>
> **W1. The FSG Analysis.** Thanks. Following prior work [3], we integrate an invariant subgraph generator into the federated learning framework. This component identifies invariant and variant subgraphs by analyzing gradient information during training. Specifically, we compute the gradients of the local loss concerning structural and attribute dimensions and use them to assign importance scores to edges. Edges with low gradient variance across clients, which reflect stable contributions to the learning objective, are selected to form the invariant subgraph $G_I$ that captures globally shared structural patterns. In contrast, edges with high gradient sensitivity are included in the variant subgraph $G_V$ to preserve client-specific information. To ensure the reliability of this partitioning, we adopt a gradient-based selection strategy controlled by a tunable threshold parameter $\varepsilon$ (refer to Eq. 12), which defines the minimum divergence required to identify variant nodes. Moreover, a rigorous theoretical analysis is presented in Section 3.4.
>
> **W2. & Q3. Implementation Details.** Thanks for your comments. We clarify that our data partitioning strictly follows established baselines to ensure fair and consistent comparisons. As downstream tasks involve varying class structures, each client is assigned a full dataset, thereby introducing strong non-IID conditions. For classification, we adopt the client-level non-IID protocol from GCFL [1]. Details regarding dataset descriptions and splits are provided in Appendix C.
>
> **W3. Time Complexity Analysis.** Thanks for your comments. The per-round and per-client computation in FedIGL mainly involves two parts: **(1)** The Federated Subgraph Generator (FSG), which consists of a $L_1$-layer GNN with complexity $O\big(L_1 (|E| d + |V| d^2)\big)$ to encode node features, and an edge-wise MLP scorer with complexity $O(|E| d^2)$for selecting invariant edges; **(2)**  The dual-branch GNN encoder, each branch having $L_2$ layers, resulting in a total complexity of $O\big(L_2 (|E| d + |V| d^2)\big)$. Here $|V|$ and $|E|$ denote the number of nodes and edges in the client's local graph, respectively; \(d\) is the feature dimensionality of each node; and $L_1, L_2$ represent the number of GNN layers in the Federated Subgraph Generator and dual-branch encoder, respectively. Hence, the overall per-client cost per round is: $O\big( (L_1 + L_2)(|E| d + |V| d^2) + |E| d^2 \big)$, which remains linear in the graph size and thus comparable to standard GNN-based federated learning methods. The extra overhead from edge scoring is lightweight and only applied once per round. In terms of communication efficiency, FedIGL only transmits partial gradients of the invariant encoder, significantly reducing bandwidth consumption compared to approaches that synchronize full models or personalized components. We will further investigate the scalability of FedIGL on larger-scale graph datasets in future work, aiming to enhance its effectiveness and practicality in real-world federated environments.
>
> **Q1. Formula Interpretation.** Thanks for your comments. In this paper, $F_I$ denotes the Invariant Feature Encoder, which is designed to extract globally shared representations from client-invariant subgraphs, thereby enhancing the generalization ability of the global model. In contrast, $F_V$ refers to the Variant Feature Encoder, which captures client-specific information to preserve local characteristics and improve the performance of personalized models. Clearer and more interpretable annotations will be provided in the final version to support better understanding.
>
> **Q2. Convergence Behavior.** We appreciate the reviewer's interest in the theoretical grounding of our method. Equation (16) formally quantifies the trade-off between convergence of the invariant encoder (refer to Proposition 3.1) and representation separation for the variant encoder (refer to Proposition 3.2), thereby establishing a convergence guarantee under the Bi-Gradient Regularization strategy. Specifically, the gradient consistency regularization $R_{inv}$ aligns the optimization directions of clients in the invariant encoder, enabling more stable training in federated settings. Meanwhile, $R_{var}$ enhances client-specific diversity by encouraging representation separation, thereby preserving local adaptability. These two regularization terms strike a balance between global invariance and local personalization. Furthermore, the learned representation is jointly optimized to ensure invariant alignment across clients via $\theta_{\text{inv}}$, while preserving per-client diversity through $\theta_{\text{var}}$. As the detailed proofs are already provided in Appendix A. Furthermore, as shown in Figure 3, our method demonstrates a smooth and monotonic reduction in the global objective across rounds, empirically validating the convergence behavior.
>
> **Q4. Writing Improvements.** Following your suggestions. All missing citations and writing issues will be addressed and corrected in the final version.
>
> [ 1 ] Federated graph classification over non-iid graphs. NeuralPS. 2021.
> [ 2 ] Federated graph-level clustering network. AAAI. 2025.
> [ 3 ] Graph Invariant Learning with Subgraph Co-mixup for Out-of-Distribution Generalization. AAAI. 2024.

---

> > ### Comment · Reviewer_hKG3 · 2025-08-03
> >
> > After reading the author's response, most of my concerns have been well addressed. I maintain my score.

---

> > > ### Author Response · Authors · 2025-08-04
> > >
> > > Dear Reviewer hKG3,
> > >
> > > We sincerely thank you for the detailed and valuable comments on our paper.
> > >
> > > Best regards,
> > >
> > > Authors of Paper 10359

---

### Official Review · Reviewer_KuhH · 2025-06-29

**Clarity:** 3
**Significance:** 4
**Originality:** 4
**Rating:** 5
**Confidence:** 4

**Summary:**

This paper presents Federated Invariant Graph Learning (FedIGL), a novel framework that effectively designed to address the challenge of non-IID data distributions in federated graph learning. Specifically, FedIGL leverages invariant learning to suppress spurious correlations and extract consistent subgraph patterns across non-iid client distributions. Moreover, a Bi-Gradient Regularization strategy is introduced to disentangle shared and client-specific representations by coordinating global and local models. Experiments on graph-level benchmarks show that FedIGL consistently improves performance on both clustering and classification tasks.

**Questions:**

Please check the weaknesses.

**Ethical Concerns:**

["NO or VERY MINOR ethics concerns only"]

**Final Justification:**

As all of my concerns have been addressed;I keep my rating of accept.

**Limitations:**

Yes

**Paper Formatting Concerns:**

No formatting concerns.

**Quality:**

3

**Strengths And Weaknesses:**

Strengths:
1. This paper addresses the limitations of data heterogeneity in federated graph learning by disentangle client-invariant and client-specific sub-graphs, offering a new approach for non-IID graph data.
2. The framework description and formula derivations are generally clear, and the flowchart aids in understanding the overall process.
3. Comprehensive experiments show statistically significant results, ablation studies and parameter analysis validate the necessity of each module.

Weaknesses:
1. While the Bi-Gradient Regularization strategy effectively separates invariant and client-specific subgraph patterns, it remains unclear how this separation aligns with the overall optimization objectives of federated learning?
2. In Equation (11), ε is introduced as a margin to enforce gradient diversity across clients. Could the authors clarify its valid range or provide guidance on how this parameter should be selected in practice?
3. In Fig. 2, the annotations for $f_V(G_V;\theta_{\text{var}})$ and $f_I(G_V; \theta_{\text{inv}})$ are missing. Providing these would improve the clarity and interpretability of the illustration.
4. A clearer description of the experimental setup would enhance transparency. In particular, it is important to clarify how the non-IID configurations differ between the classification and clustering tasks. What motivates these design choices, and how are the datasets partitioned across clients in each case?
5. The scalability analysis with respect to the number of clients is insufficient. How does the proposed framework perform as the number of clients increases?
6. A more comprehensive discussion of privacy is needed. Could the sharing of gradients during training lead to potential information leakage? Moreover, how does the proposed method compare to existing privacy-preserving baselines in terms of its ability to protect sensitive client information?

---

> ### Author Rebuttal · Authors · 2025-07-28
>
> **Q1. Clarification of the Optimization Objective.** We appreciate your comment. Most existing works achieve this by decomposing the global objective into a weighted sum of local empirical losses and performing independent optimization on each client. However, under non-IID conditions, local updates from different clients may follow divergent optimization directions, impairing convergence and model generalization. Our proposed Bi-Gradient Regularization strategy facilitates decoupled and efficient feature learning across distributed clients to optimize the global model objective. Specifically, we first employ the FSG module to disentangle client-agnostic and client-specific subgraphs. Then, the gradient alignment mechanism minimizes the variance of gradients across clients on the invariant encoder, promoting consistency in the shared subgraph representation and guiding the global model toward a unified optimization direction. In contrast, the diversity of gradients on the variant encoder preserves client-specific structural information, preventing the dilution of personalized knowledge during aggregation. Moreover, FedIGL effectively fosters collaboration among clients to uncover invariant subgraph patterns across heterogeneous distributions, thereby mitigating client drift induced by structural heterogeneity.
>
> **Q2. Sensitivity Analysis of ε.** We appreciate your comment. In our work, ε is introduced as a lower bound on the gradient differences in the variant branch across clients, encouraging meaningful diversity in learning directions and thereby preserving client-specific personalized features. We tested ε values in the range from 0.1 to 0.3 and found that they yielded the best performance, as shown in Fig. 4 of the manuscript.
>
> **Q3.  Better Understanding.** Thank you for your suggestion. In our framework diagram, $f_V(G_V; \theta_{\text{var}})$ and $f_I(G_V; \theta_{\text{inv}})$ represent the variant and invariant subgraph encoders, respectively. We will enhance the clarity of this notation and include corresponding explanations in the figure caption of the final version.
>
> **Q4. Implementation Details.** Thanks for your comments. (1) The settings are motivated by real-world federated scenarios, where clients naturally hold heterogeneous data due to differences in data sources, tasks, or domains. This setup also aligns with widely adopted benchmarks [1–2]. (2) In both classification and clustering tasks, each client is assigned a distinct dataset to simulate highly non-IID settings.  In classification tasks, the non-IID nature primarily stems from differences in the number of classes, label distributions, and domain semantics across client datasets. In clustering tasks, non-IIDness arises from variations in the number of clusters, clustering structures, and domain-specific graph properties.
>
> **Q5. Scalability Analysis.** Thank you for your valuable comments. We appreciate the reviewer’s suggestion regarding scalability. To evaluate the scalability of our proposed FedIGL framework, we conducted additional experiments on the SM dataset by varying the number of participating clients while keeping the total dataset size fixed. Specifically, we tested configurations with 5, 10, and 20 clients under consistent non-IID conditions. As summarized in Table 1, FedIGL achieves stable accuracy across various client numbers. Although performance slightly decreases as the number of clients increases, the overall results remain stable and reliable.
> ### Table 1
> | Number of Clients |   5   |   10   |   20   |
> |-------------------|-------|--------|--------|
> | ACC ( % )           | 83.07 ± 1.76 | 83.11 ± 1.06  | 82.78 ± 0.97  |
>
> **Q6. Privacy Analysis.** Thanks. Compared to existing FGL methods that share global prototypes or the full set of encoder parameters, FedIGL transmits only the gradients of partial invariant components. More precisely, the client-specific subgraph structures remain entirely local and are never involved in communication. Moreover, the shared invariant subgraph encoder captures structural patterns that are highly similar across clients. Under non-IID conditions, such partial and redundant information makes it difficult to reconstruct a client’s complete local graph. This design helps reduce potential privacy leakage risks while preserving effective collaboration. We further evaluate the utility of FedIGL under differential privacy by injecting noise into model updates on the SM dataset, across a range of privacy budgets $\varepsilon$ $\in$ {$\infty$, 3, 1, 0.5, 0.1}. As shown in Table 2, FedIGL consistently achieves the highest accuracy under all privacy levels, maintaining 81.47% even at the strictest setting of $\varepsilon = 0.1$, and significantly outperforming both GCFL [2] and FedStar [3]. Empirical evidence indicates that FedIGL maintains strong model performance despite operating under strict privacy constraints.
> ### Table 2
> |     Methond    | GCFL | FedStar | FedIGL |
> |:-:|:-:|:-:|:-:|
> |  ACC （ε = $\infty$)   |   77.61 ± 1.79    |     75.39 ± 2.03  |    83.07 ± 1.76        |
> | ACC (w/ DP, $\varepsilon$=3) |   75.78 ± 1.30  | 72.16 ± 1.67    |    82.09 ± 2.03  |
> | ACC (w/ DP, $\varepsilon$=1) |   75.61 ± 2.19  |71.49 ± 1.12     |    81.32 ± 1.19      |
> | ACC (w/ DP, $\varepsilon$=0.5) |   74.17 ± 1.41   | 72.02 ± 0.89     |   81.81 ± 1.03      |
> | ACC (w/ DP, $\varepsilon$=0.1) |   74.08 ± 3.02  | 71.75  ± 1.99     |    81.47 ± 1.35      |
>
> [ 1 ] Federated graph classification over non-iid graphs. NeuralPS, 2021.
> [ 2 ] Federated graph-level clustering network. AAAI, 2025.
> [ 3 ] Federated learning on non-iid graphs via structural knowledge sharing. AAAI, 2023.
> [ 4 ] Privacy-Preserving Personalized Federated Prompt Learning for Multimodal Large Language Models. ICLR, 2025.

---

> > ### Comment · Reviewer_KuhH · 2025-08-07
> >
> > Thank you for your detailed response. All of my concerns have been addressed; I keep my rating of accept.

---

### Official Review · Reviewer_n8up · 2025-06-30

**Clarity:** 3
**Significance:** 4
**Originality:** 3
**Rating:** 4
**Confidence:** 5

**Summary:**

This work develops a novel framework called FedIGL, aimed at mitigating data heterogeneity in federated graph learning. Rather than relying on statistical aggregation of structural information, the proposed approach employs a Bi-Gradient Regularization strategy that enables clients to effectively disentangle invariant and client-specific representations. Experimental results demonstrate that FedIGL exhibits promising performance across various graph datasets.

**Questions:**

1. Eq. (12) lacks sufficient detail regarding the optimization of losses. A more explanation of their optimization would improve clarity and reproducibility.
2. How does the proposed framework protect privacy in both the global and local models?
3. What do the metrics in Table 1 represent?
4. How the heterogeneous data distribution are defined for both clustering and classification？
5. There a missing reference at line 83.

**Ethical Concerns:**

["NO or VERY MINOR ethics concerns only"]

**Final Justification:**

I thank the authors for their detailed rebuttal. I believe my concerns have been well addressed, and their clarifications have improved my understanding and appreciation of this work, so I will keep my positive rating.

**Limitations:**

Yes

**Quality:**

3

**Strengths And Weaknesses:**

Strengths
1. The paper offers an interesting perspective and appears to be well-executed. The methodology is both rigorous and novel, with each module clearly and intuitively explained. The model design effectively emphasizes invariant subgraph patterns in addressing structural heterogeneity in federated graph learning.
2. Extensive experiments and theoretical analysis strengthen the empirical findings, suggesting that FedIGL exhibits strong potential and competitive performance in relevant tasks.

Weaknesses
1. While the FSG module appears well-designed, further clarification is needed on how its separation mechanism aligns with the learning objectives of downstream tasks in federated settings. In particular, it remains unclear how the global and local loss functions are coordinated within the proposed framework to ensure consistent and effective optimization.
2. The explanation of the experimental setup is insufficient, particularly with respect to how labels are partitioned in classification tasks and how heterogeneous data distributions are defined and constructed in both clustering and classification settings.
3. While the use of subgraph gradients may offer implicit privacy benefits, the lack of a formal privacy analysis limits the validity of comparisons with established privacy-preserving techniques. The authors are encouraged to provide a rigorous privacy evaluation to strengthen their claims.

---

> ### Author Rebuttal · Authors · 2025-07-29
>
> **W1. Global and Local Loss.** Thank you for your comments. The objective of standard Federated Learning methods is to learn a global model that minimizes the empirical risk across all client distributions. FedIGL achieves consistent and effective optimization by systematically coordinating global and local loss functions through structural disentanglement and bi-gradient regularization. Specifically, each input graph is partitioned into a client-agnostic invariant subgraph and a client-specific variant subgraph via the proposed FSG module. The global loss is imposed on the invariant representation, which is optimized through cross-client aggregation, with the consistency regularizer $R_{inv}$ reducing gradient variance across clients to ensure convergence to a consistent global objective (refer to Proposition 3.1, Section 3.4). Meanwhile, the local loss is computed solely on the client-specific subgraph, with its encoder remaining strictly local and personalized. The diversity regularizer $R_{var}$ encourages representational separation per-client (refer to Proposition 3.2), thereby preserving local adaptation. This design allows the invariant encoder to achieve global alignment while the variant encoder supports personalization, ensuring stable and collaborative optimization in structurally heterogeneous federated graph learning.
>
> **W2. Confusing in classification tasks.** Thanks. Following prior work[1], each client is assigned a separate dataset. In intra-domain scenarios, clients are associated with different datasets within the same domain, each exhibiting distinct class priors or label distributions, thereby introducing label-level heterogeneity. In cross-domain scenarios, clients are assigned datasets from entirely different domains (e.g., small molecules, bioinformatics, and computer vision), where substantial differences exist in label semantics and structural characteristics. In each of the settings, a client owns one of the corresponding datasets and randomly splits it into three parts: 80% for training, 10% for validation, and 10% for testing.
>
> **W3. & Q2. Privacy Analysis.** Thanks. The architecture of FedIGL is intrinsically aligned with privacy-preserving objectives. All training operations, including updates to both the invariant and variant encoders, are executed locally on the client side. The variant encoder is strictly confined to the client and remains entirely excluded from any communication. Moreover, the gradients shared during training pertain only to the invariant subgraph encoder, which captures structurally similar patterns across clients. The limited informativeness of shared gradients under non-IID settings substantially reduces the risk of reconstructing client-specific graph data. In contrast to methods that transmit prototypes, full model parameters, or complete client updates, FedIGL restricts the shared content to structurally coarse and partially informative components, thereby substantially narrowing the attack surface and enhancing resilience to gradient-based privacy attacks. We further evaluate the utility of FedIGL under differential privacy by injecting noise into the model on the SM dataset, across a range of privacy budgets $\varepsilon \in {\infty, 3, 1, 0.5, 0.1}$. As shown in Table 1, FedIGL consistently achieves the highest accuracy under all privacy levels, maintaining 81.47% even at the strictest setting of $\varepsilon = 0.1$, and significantly outperforming both GCFL and FedStar [3]. These results confirm that FedIGL preserves high model utility even under stringent privacy constraints.
> ### Table 1
> |     Methond    | GCFL | FedStar | FedIGL |
> |:-:|:-:|:-:|:-:|
> |  ACC （ε = $\infty$)   |   77.61 ± 1.79    |     75.39 ± 2.03  |    83.07 ± 1.76        |
> | ACC (w/ DP, $\varepsilon$=3) |   75.78 ± 1.30  | 72.16 ± 1.67    |    82.09 ± 2.03  |
> | ACC (w/ DP, $\varepsilon$=1) |   75.61 ± 2.19  |71.49 ± 1.12     |    81.32 ± 1.19      |
> | ACC (w/ DP, $\varepsilon$=0.5) |   74.17 ± 1.41   | 72.02 ± 0.89     |   81.81 ± 1.03      |
> | ACC (w/ DP, $\varepsilon$=0.1) |   74.08 ± 3.02  | 71.75  ± 1.99     |    81.47 ± 1.35      |
>
> **Q1. Confusing in Eq.(12).** Thanks. Eq. (12) defines a bi-gradient regularization objective for global model aggregation. It is important to clarify that the local training loss on each client is not omitted. Specifically, in each communication round, the server compares the gradients uploaded by clients with the previously aggregated global gradients. The gradients of the invariant branch are regularized to be consistent across clients, promoting the generalization of the global model, while the variant branch gradients are encouraged to diverge across clients, preserving personalization for local models. Meanwhile, each client optimizes its task-specific loss on local data to update the variant encoder, while the global model integrates consistency and diversity regularizers to guide distributed representation learning, fostering synergy between local personalization and global generalization.
>
> **Q3. & Q5. Improving Writing.** Thank you for your suggestion. The metric reported in Table 1 is accuracy, and we will explicitly clarify this in the table caption in the final version of the manuscript. Furthermore, we will ensure that all missing citations are properly added and will carefully revise the manuscript to correct any remaining inconsistencies in metric reporting and formatting.
>
> **Q4. Definition of Heterogeneous Data Distributions.** Thanks. We follow the experimental settings established in prior work [1, 2], utilizing datasets from diverse domains such as small molecules, bioinformatics, and computer vision. These datasets are used to construct heterogeneous client distributions that reflect both label distribution shifts, classes, and domain-level variations. For both classification and clustering tasks, each client is assigned a distinct partition of the data. In single-domain scenarios, clients are sampled from different datasets within the same domain, introducing heterogeneity in label spaces or clustering structures. In cross-domain settings, clients originate from entirely different domains, thereby exhibiting significant disparities in label semantics, feature distributions, and underlying graph topologies. The non-IID settings and dataset details for both classification and clustering tasks are provided in Appendix C.
>
> [ 1 ] Federated graph classification over non-iid graphs, NeuraIPS, 2021.
> [ 2 ] Federated graph-level clustering network, AAAI, 2025.
> [ 3 ] Federated learning on non-iid graphs via structural knowledge sharing, AAAI, 2023.

---

> ### Comment · Reviewer_n8up · 2025-08-03
>
> I thank the authors for their detailed rebuttal. I believe my concerns have been well addressed, and their clarifications have improved my understanding and appreciation of this work, so I will keep my positive rating.

---

> > ### Author Response · Authors · 2025-08-04
> >
> > Dear Reviewer n8up,
> >
> > We sincerely thank you for the detailed and valuable comments on our paper.
> >
> > Best regards,
> >
> > Authors of Paper 10359

---

### Official Review · Reviewer_DuEb · 2025-07-05

**Clarity:** 3
**Significance:** 3
**Originality:** 2
**Rating:** 4
**Confidence:** 4

**Summary:**

The paper introduces FedIGL, a federated graph-learning framework that aims to improve generalization under non-IID client distributions by (i) splitting each client graph into “invariant” and “variant” subgraphs with a federated subgraph generator, and (ii) applying a Bi-Gradient Regularization (gradient alignment for invariant features, gradient repulsion for variant features).

**Questions:**

1.  How sensitive is FedIGL to early-round noise? Would a warm-up or EMA of historical gradients improve stability?

1. Please specify exactly how graphs were split into clients (by dataset, by class, or random). Performance can vary drastically with partition severity.

1. Given recent gradient-inversion attacks on GNNs, can the authors quantify information leakage under their gradient-sharing scheme? Differential-privacy noise or secure aggregation would strengthen the story.

**Ethical Concerns:**

["NO or VERY MINOR ethics concerns only"]

**Limitations:**

yes

**Quality:**

3

**Strengths And Weaknesses:**

Strengths：
1. End-to-end implementation with a reasonable experimental breadth (19 datasets).
1. High-level idea is easy to grasp; figures help.
1. First explicit pairing of gradient alignment/repulsion within a subgraph-mask framework for FGL.

Weaknesses：
1. Baseline selection omits strong personalized-FL and IRM baselines (FedDyn, Ditto, FedRep, IRM-GNN).
1. Key hyper-parameters (client splits, local epochs, comm. cost) undocumented, hurting reproducibility.
1. Incremental combination of known ideas (IRM-style alignment + gradient diversity) yields limited novelty; impact likely modest without stronger theory or privacy guarantees.

---

> ### Author Rebuttal · Authors · 2025-07-29
>
> **W1. More Baseline.** Thank your suggestions. To ensure fair comparison, we benchmarked against SOTA federated graph learning methods. Following your suggestion, we additionally conducted experiments with the referenced baselines on the SM and SM-BIO datasets. Table 1 demonstrates that FedIGL achieves significantly better performance than all baselines across both classification and clustering tasks. Note that GIL [1] is IRM-GNN without a federated framework; we apply FedAvg for parameter aggregation to ensure a consistent evaluation setup.
>
> ### Table 1
> | Dataset | FedDyn | Ditto  | FedRep | GIL* | OURS   |
> |-|-|-|-|-|-|
> | SM | 76.87 ± 1.54   | 72.14 ± 01.87  | 73.43 ± 0.85  | 73.75 ± 2.09  | 83.07 ± 1.76 |
> | SM-BIO | 70.87 ± 2.14  | 67.48 ± 2.05   | 65.67 ± 1.77   | 71.12 ± 1.40   | 77.02 ± 1.32 |
>
> [ 1 ] Learning Invariant Graph Representations for Out-of-Distribution Generalization. Neurips, 2022.
>
> **W2. & Q2. Experimental Details.** Thanks for your suggestions. We would like to clarify that our data partition protocols strictly follow standard baselines to ensure fair comparison. Specifically, for the classification tasks, we adopt the client-level non-IID partition from GCFL [2], and for clustering tasks, we follow FedGCN[3] Details of dataset splits and configurations can be found in Appendix C.1. Specifically, **Dataset Partition:** As downstream tasks differ in the number of classes, we assign a complete dataset to each client, naturally introducing a high level of non-IIDness. **Local Epochs:** Following prior work, each client is trained for 500 local epochs. We also provide a convergence analysis, shown in Fig.3. **comm. cost:** On the SM dataset with 7 clients, FedIGL incurs 0.2315 MB of communication overhead per client per round, compared to 0.2643 MB for GCFL, indicating a moderate reduction in communication cost. In summary, more details of the experimental setup will be provided in the final version.
>
> [ 2 ] Federated graph classification over non-iid graphs. NeuraIPS, 2021.
> [ 3 ] Federated graph-level clustering network. AAAI, 2025.
>
> **W3. Technical Contributions.** Thank your comments. We would like to further clarify the novelty and technical contributions of our work. Our main contributions can be summarized as follows:
> **(i) Novel Perspective.** Prior works that impose overly strict assumptions about the presumed correlation between structural features and the global objective often fail to generalize to local tasks.  Our work focuses on identifying and leveraging client-invariant, discriminative factors to reduce the effect of spurious correlations. To the best of our knowledge, this is the first work to introduce invariant subgraph structures into FGL to address the distribution shifts challenge.
> **(ii) Technical Contributions and Theoretical Analysis.** We propose a bi-gradient regularization framework that fundamentally advances beyond the classical IRM paradigm and directly addresses the optimization challenges of federated learning under non-IID distributions. Unlike existing methods that merely alleviate sub-optimal results caused by gradient heterogeneity, our approach introduces a structural disentanglement strategy that explicitly decomposes each graph into invariant and variant subgraphs. These are jointly optimized via two complementary regularizers: gradient consistency ensures global alignment, while gradient diversity drives personalized adaptation. This design not only mitigates conflicts arising from structural heterogeneity but also achieves collaborative optimization with enhanced personalization, marking a clear step forward in federated graph learning.
> We provide a theoretical analysis in Section 3.4 to formally support the proposed mechanism. Specifically, we prove that the introduced regularizers contribute to the convergence of the global optimization objective and effectively enhance the discriminability of local structural representations. We have provided detailed theoretical proofs in Appendix A.
> **(iii) Extensive Experiments and Privacy Preservation.** In contrast to most existing methods that are typically validated on a single task, we conduct extensive experiments on both classification and clustering tasks under non-IID settings. Results against advanced baselines validate the generalizability of FedIGL.Additionally, unlike prior approaches that commonly share global prototypes or full structural representations, potentially exposing sensitive local information, FedIGL shares only a subset of subgraph structures with high structural similarity across clients. Such information is inherently less reconstructable, offering a promising degree of privacy protection. Furthermore, we provide additional experimental validation in our response to Question 3.
> In summary, FedIGL prioritizes simplicity and efficiency over overly complex design.
>
> **Q1. Training Setup.** To ensure fair comparison, our training protocol strictly follows prior works, focusing instead on the core framework design. We conducted additional experiments measuring early-round accuracy variance (T = 5, 10) in SM datasets. Table 2 shows that FedIGL exhibits low accuracy variance at both round 5 and round 10, with only minor fluctuations, indicating strong training stability. Furthermore, the incorporation of EMA yields only marginal improvements in both variance and accuracy, suggesting that FedIGL is inherently stable, even without additional smoothing techniques.
>
> ### Table 2
> | Method       | Variance (T=5) | Variance (T=10) | ACC (%) |
> |--------------|----------------|------------------|---------|
> | FedIGL       | 5.83         | 5.29           | 83.07 ± 1.76   |
> | FedIGL+EMA   | 4.91         | 4.56          | 84.15 ± 1.20   |
>
> **Q3. Privacy Analysis.**
> We conduct additional experiments to evaluate the privacy-preserving capability of FedIGL, incorporating both gradient inversion attacks via DLG (Deep Leakage from Gradients) [4,5] and formal privacy guarantees through DP [6].
> **(i)** We employ DLG to reconstruct node features from shared gradients and use the MSE between the original and reconstructed features as a proxy for privacy leakage. As reported in Table 3, FedIGL achieves an MSE of 0.60 on SM and 0.52 on BZR, which are slightly higher than those of the strongest baselines. Given that all features are normalized to the [0,1] range, these values suggest that attackers face substantial difficulty in accurately recovering the original inputs. Furthermore, unlike in vision domains, graph data comprises both node attributes and topological structure, and the discrete nature of graph connectivity further increases the difficulty of faithful reconstruction.
>
> ### Table 3
> | Dataset | FedIGL (+DLG) | GCFL (+DLG) |
> |:--------|:--------------:|:-----------:|
> | SM      |     0.63       |    0.59     |
> | BZR     |     0.52       |    0.50     |
>
> **(ii)** We further evaluate the performance of FedIGL on the SM dataset by injecting differential privacy noise under varying privacy budgets $\varepsilon$ $\in$ {$\infty$, 3, 1, 0.5, 0.1}. As shown in Table 4, FedIGL consistently achieves the highest accuracy under all privacy settings, reaching 81.47% even at the strictest budget of $\varepsilon$ = 0.1, and significantly outperforming competing methods. These results indicate that FedIGL effectively mitigates the performance degradation caused by differential privacy and demonstrates strong privacy-preserving capability.
> ### Table 4
> |     Methond    | GCFL | FedStar | FedIGL |
> |:-:|:-:|:-:|:-:|
> |  ACC （ε = $\infty$)   |   77.61 ± 1.79    |     75.39 ± 2.03  |    83.07 ± 1.76        |
> | ACC (w/ DP, $\varepsilon$=3) |   75.78 ± 1.30  | 72.16 ± 1.67    |    82.09 ± 2.03  |
> | ACC (w/ DP, $\varepsilon$=1) |   75.61 ± 2.19  |71.49 ± 1.12     |    81.32 ± 1.19      |
> | ACC (w/ DP, $\varepsilon$=0.5) |   74.17 ± 1.41   | 72.02 ± 0.89     |   81.81 ± 1.03      |
> | ACC (w/ DP, $\varepsilon$=0.1) |   74.08 ± 3.02  | 71.75  ± 1.99     |    81.47 ± 1.35      |
>
> These results demonstrate the privacy-preserving potential of FedIGL, which can be attributed to the proposed bi-gradient regularization mechanism that limits shared information to invariant substructure encoders, thereby avoiding the exposure of complete graph structures or gradients.
>
> [ 4 ] Deep Leakage from Gradient. NeurIPS, 2019.
> [ 5 ] Hiding in Plain Sight: Disguising Data Stealing Attacks in Federated Learning.  ICLR,  2024.
> [ 6 ] Privacy-Preserving Personalized Federated Prompt Learning for Multimodal Large Language Models. ICLR, 2025.

---

### Note · Authors · 2025-08-16

Dear Area Chair and Reviewers,

We sincerely appreciate your valuable and constructive comments, with most recognizing the innovation and value of this study and giving positive scores. We would further summarize the contributions of this study and the concerns addressed in the rebuttal.

**Technical Contributions:** We clarify the innovation concern raised by Reviewer **DuEb**. Prior works that impose overly strict assumptions about the presumed correlation between structural features and the global objective often fail to generalize to local tasks. By contrast, we center our innovation on decouple client-invariant and client-specific subgraphs via FSG and propose a bi-gradient regularization, where gradient consistency ensures global alignment and gradient diversity drives personalized adaptation. Our approach simultaneously exploits discriminative factors to alleviate the impact of spurious correlations and strengthens privacy preservation.

**Theoretical Analysis:** We present a detailed analysis of the bi-gradient regularization and the associated loss functions to respond to the comments from Reviewers **n8up** and **KuhH**. Additionally, we provide a time complexity analysis to address the concern by Reviewer **hKG3**, thereby demonstrating the theoretical soundness and practical efficiency of our method.

**Experimental Validation:** We conduct extensive additional experiments, including the differential privacy experiments suggested by Reviewers **KuhH**, **n8up**, and **DuEb**; the comparison with personalized-FL and IRM baselines, the analysis of FedIGL’s sensitivity and stability under early-round noise, and the evaluation of its performance under DLG (Deep Leakage from Gradients) attacks as requested by Reviewer **DuEb**; the scalability analysis emphasized by Reviewer **KuhH**. These experiments further validate the theoretical soundness and practical efficiency of our method.

The positive comments from Reviewers **n8up**, **DuEb**, **KuhH**, and **hKG3** have acknowledged the technical quality and significance of our work. We believe our detailed responses have comprehensively addressed all concerns and fully demonstrated the merit of our method.

Best regards,
Authors of Paper with ID 10359

---

### Decision · Program_Chairs · 2025-09-17

**Decision:**

Accept (poster)

**Comment:**

The authors present Federated Invariant Graph Learning (FedIGL), a framework that combines a global model for collaborative training with client-specific models to capture both shared and local subgraph patterns. They also introduce a Bi-Gradient Regularization strategy to disentangle invariant and spurious correlations, resulting in superior performance on graph clustering and classification tasks.

This paper received generally positive reviews, with final scores ranging from borderline accept to accept. The authors made a strong effort during the rebuttal to address reviewers’ concerns, and most reviewers acknowledged the responses as satisfactory (though reviewer DuEb did not provide a final justification, his/her initial weak accept score was already positive).

Overall, the AC considers the paper suitable for acceptance, given the favorable consensus and effective resolution of reviewer feedback.